

# Deep learning models for damage type detection in wind turbines

Ferdi Doğan[1], Saadin Oyucu[2], Emre Bicer[3] and Ahmet Aksoz[4]

[1] Computer Engineer, University of Adıyaman, Adıyaman, Turkey
[2] Department of Computer Engineering/Faculty of Technology, Gazi University, Ankara, Turkey
[3] Battery Research Laboratory, Sivas University of Science and Technology, Sivas, Turkey
[4] Mobilers Team, Sivas Cumhuriyet University, Sivas, Turkey

## ABSTRACT

This study presents deep learning models that are frequently used in the literature for the detection and classification of damage types in wind turbines and a new deep learning model (SatNET) that offers computational efficiency and rapid inference. Wind turbines, which are critical components of renewable energy systems, are sensitive to various damages (paint damage, erosion, serration, vortex, and vortex damage) that may endanger their operational efficiency and lifespan. The dataset consists of 1,794 high-resolution images taken under different weather conditions and angles, including damage and types. The images were increased by four times to 7,176 images using data augmentation techniques. Damage and types were detected using the developed SatNET deep learning model, 11 deep learning models, and the Faster Region-based Convulational Neural Network (R-CNN) object detection algorithm. Each of the models was evaluated with average sensitivity. Accordingly, SatNET achieved avarage precision (AP) values of 55.7% for paint damage, 76.7% for erosion, 95.2% for serration, 66.1% for vortex, and 27.3% for vortex damage. It demonstrated superior performance when compared to deep learning models frequently used in the literature, such as ResNet50 and VGG19. In addition, it has been shown that the model requires less computational cost than other models, with a memory requirement of 192 MB. The results show that SatNET's computational efficiency and accuracy are competitive with other models. The model is suitable for systems with limited memory and computational capacity, which require real-time operation, and for systems with resource constraints. The results obtained can contribute to sustainability in renewable energy production by providing low-cost monitoring of damage and types in wind turbines.

## INTRODUCTION

Renewable energy sources, such as solar, wind, hydro, and geothermal, play a crucial role in addressing the global energy crisis and combating climate change. Unlike fossil fuels, which are finite and emit greenhouse gases, renewable energy sources are sustainable and have a minimal environmental impact (*Bhuiyan, 2022*). Renewable energy reduces dependency on imported fuels, enhances energy security, and creates jobs in emerging green technologies. Furthermore, transitioning to renewable energy contributes to cleaner

Corresponding author
Ferdi Doğan,
fdogan@adiyaman.edu.tr

air and water, improving public health and quality of life. As the world seeks to meet rising energy demands while mitigating climate change effects, investing in renewable energy is essential for a sustainable future (*Haines et al., 2007*).

Wind turbines provide a clean and sustainable alternative to fossil fuels. However, they are susceptible to various damages that can significantly impact their efficiency and lifespan. Key types include paint damage, which leads to corrosion and increases maintenance costs; erosion, which reduces aerodynamic efficiency and energy output; serration, which causes vibrations that can hinder performance; vortex, where airflow disruptions decrease energy generation; and vortex damage, which can result in structural harm to the blades. Regular monitoring and maintenance of wind turbines are essential to detect these issues early, ensuring optimal performance and extending their operational life while maintaining energy sustainability (*González-Salcedo et al., 2020*; *Katsaprakakis, Papadakis & Ntintakis, 2021*). Detecting damage and contamination in wind turbines is essential for maintaining their efficiency and extending their lifespan. If damage and contamination are not identified in time, significant energy production losses can occur, and maintenance costs may rise. Blade wear, paint damage, and erosion substantially degrade of turbines, leading to energy losses. Therefore, modern wind turbines are regularly monitored using visual inspections, drone-based assessments, and sensor-based monitoring systems. Early detection of damage enables the optimization of scheduled maintenance processes, extends the turbine's operational life, and minimizes downtime caused by failures. These approaches help reduce operational costs while ensuring the stability of sustainable energy production (*Matani, 2021*; *Taraglio et al., 2024*).

Image processing technology and deep learning play a crucial role in the detection and analysis of damage in wind turbines and other industrial systems. Advanced image processing techniques can analyze high-resolution images or video data collected from turbines to detect surface defects like erosion, cracks, or paint damage. Deep learning algorithms, especially convolutional neural networks (CNNs), can automatically classify and detect these damages with high accuracy, even in challenging environments. By training these models on large datasets of turbine images, they can identify subtle signs of wear or contamination that may be missed during manual inspections. The integration of image processing and deep learning not only increases the speed and accuracy of damage detection but also reduces the need for costly and time-consuming manual inspections, improving maintenance efficiency and operational reliability (*Guo et al., 2021*; *Wang et al., 2022*).

The detection of damage and contamination in wind turbines using traditional methods is typically conducted through manual inspections, periodic audits, and physical examinations. These processes require maintenance teams to stop the turbines and visually inspect them, particularly the blades and other critical components. In traditional methods, technical personnel attempt to identify paint damage, erosion, cracks, and other types of contamination on the turbine blades using cranes, drones, or telescopic cameras. However, these methods can be time-consuming, labor-intensive, and costly. Furthermore, these inspections are often limited by the constraints of human eyesight and experience,

which can result in small damages or early-stage wear being overlooked (*Zhang & Shu, 2024*).

Deep learning models have demonstrated significant successes in the detection of damage and contamination in wind turbines, revolutionizing traditional inspection methods. These models, particularly CNNs, have been trained on large datasets of turbine images, enabling them to recognize intricate patterns associated with various types of damage, such as erosion, paint damage, cracks, and dirt accumulation. The high accuracy and reliability of these models allow for early detection of issues that might otherwise go unnoticed in manual inspections (*Liu, Hajj & Bao, 2022*).

One notable achievement is the ability of deep learning models to analyze images captured by drones or fixed cameras in real time, providing immediate feedback to maintenance teams. This capability enhances decision-making processes regarding necessary repairs and maintenance schedules, ultimately leading to reduced downtime and operational costs. Moreover, deep learning models can adapt and improve over time by incorporating new data, making them increasingly effective at identifying emerging damage types and trends (*Ren et al., 2021*).

Additionally, the application of deep learning in damage detection contributes to safer working conditions for inspection teams, as it reduces the need for personnel to perform inspections at great heights or in hazardous environments. Overall, the advancements in deep learning models not only improve the efficiency and accuracy of damage detection in wind turbines but also support the goal of maintaining high energy production reliability in renewable energy systems (*Shihavuddin et al., 2021*).

The use of artificial intelligence-based CNNs and deep learning models to detect damages in wind turbines has emerged as a groundbreaking approach in predictive maintenance. These advanced models leverage vast amounts of data from various sources, including images captured by drones and surveillance cameras, to identify and analyze potential damage such as erosion, paint wear, cracks, and contamination on turbine blades (*Memari et al., 2024*).

This study aims to detect damage and its types in wind turbines, renewable energy sources, using deep learning techniques and the model we developed. By leveraging a dataset of approximately 1,794 segmented images, including turbine blades and towers with documented damages and contamination, we strived to ensure that our model produces results comparable to or superior to existing models. The training process utilized 80% of the dataset, with 10% allocated for validation and the remaining 10% for testing, ensuring a robust evaluation of the model's performance. Through rigorous testing and assessment metrics, we aim to create a new deep learning model that effectively identifies damage types in wind turbines and contributes to enhancing the efficiency and reliability of renewable energy systems.

Among the CNN-based architectures, models such as AlexNet, VGG, ResNet, and Inception have shown significant effectiveness in image classification and object detection problems. However, these models have high computational cost. They are not optimized for embedded systems or resource-constrained systems such as drones used in wind turbine inspections. Moreover, their general-purpose designs may not capture the

fine-grained features specific to wind turbine surface damages. Considering these limitations, we propose SatNET, which is specifically designed to provide an accurate solution with low computational cost to detect various damage types in wind turbine images.

## LITERATURE REVIEW

Detecting pollution and damage in wind turbines using three main approaches: classical methods, traditional image processing techniques, and deep learning approaches. Classical methods rely on traditional techniques such as visual inspection, vibration analysis, acoustic emission monitoring, and oil analysis. Visual inspections identify surface contamination, corrosion, and structural damage on turbine components, while vibration analysis detects imbalances, misalignments, or defects in rotating machinery like gearboxes and bearings by monitoring changes in vibration patterns (*Du et al., 2020*). Acoustic emission monitoring captures high-frequency sounds generated by crack growth, friction, or other internal damage, which is especially useful for inaccessible components (*Tchakoua et al., 2014*). Oil analysis examines lubricating oil for contaminants, wear particles, and chemical degradation, providing insights into the condition of gearboxes and bearings (*Tiboni et al., 2022*). These classical approaches are reliable but are often labor-intensive and influenced by environmental conditions.

Traditional image processing methods have enhanced wind turbine monitoring by utilizing computer vision algorithms to analyze images captured by cameras or drones. Edge detection techniques, such as Sobel, Canny, and Prewitt, identify boundaries of cracks or scratches, while thresholding techniques segment images based on pixel intensity to isolate defects or contaminants (*Elforjani & Bechhoefer, 2018*). Template matching compares real-time images with reference templates to detect specific damage types, such as bolt loosening or debris accumulation (*Sharma, Ansari & Kumar, 2017*). Morphological operations, including dilation, erosion, opening, and closing, refine image features to highlight cracks, corrosion, or surface defects (*Abd Elaziz, Bhattacharyya & Lu, 2019*), while color segmentation using models like RGB and HSV differentiates between clean and polluted turbine surfaces (*Nagata et al., 2019*). Although these methods are effective, their performance can be influenced by image quality, lighting, and environmental factors, limiting their robustness under varying conditions.

Deep learning techniques, particularly CNNs, have revolutionized defect detection in wind turbines by automating the process and improving accuracy. CNNs are employed for tasks such as image classification, object detection, and semantic segmentation. Pre-trained networks like VGG, ResNet, and AlexNet are fine-tuned to classify turbine components as "damaged" or "undamaged," leveraging labeled datasets to identify various defects, including cracks and corrosion (*Chanda, 2008*). Object detection frameworks such as You Only Look Once (YOLO) and Faster Region-based Convulational Neural Network (R-CNN) detect specific defects by drawing bounding boxes around areas of interest (*Jain & Laxmi, 2018*), while segmentation models like U-Net and SegNet classify image pixels for detailed analysis of defect shapes and sizes (*Choeda & Pruthi, 2022*). Transfer learning enhances model performance when labeled datasets are limited, using weights from

**Table 1 Comparison of the performance of different deep learning models used for detecting pollution and damage in wind turbines, based on key metrics such as accuracy, precision, recall, and computational complexity.**

| Model | Task | Accuracy (%) | Precision (%) | Recall (%) | Computational complexity | Strengths | Weaknesses |
|---|---|---|---|---|---|---|---|
| ResNet (*Choeda & Pruthi, 2022*) | Image classification | 90–95 | 88–94 | 87–92 | High (Deep architecture) | High accuracy, can handle complex patterns | Requires large datasets and computational power |
| YOLOv5 (*Khanam et al., 2024*) | Object detection | 85–92 | 83–90 | 82–89 | Moderate (Real-time detection) | Fast detection, suitable for real-time applications | May miss small or subtle defects |
| Mask R-CNN (*Adhikari et al., 2024*) | Instance segmentation | 87–93 | 85–92 | 84–90 | High (Region-based network) | Pixel-level segmentation, detailed defect detection | Slower compared to YOLO due to segmentation complexity |
| LSTM (*Liu et al., 2021*) | Time-series anomaly detection | 80–88 | 78–85 | 79–87 | Moderate (Sequential data processing) | Effective for temporal data, capture time dependencies | Limited to time-series analysis, requires preprocessed data |
| U-Net (*Hao et al., 2023*) | Semantic segmentation | 88–94 | 86–91 | 85–93 | Moderate (Encoder-decoder architecture) | High accuracy in segmentation tasks | Sensitive to noise and small dataset size |
| GAN + CNN (Hybrid) (*Hao et al., 2023*) | Synthetic data generation + detection | 85–93 | 83–90 | 84–91 | High (Generative + discriminative) | Data augmentation improves model performance | Requires substantial training time |
| DeepLabV3+ (*Ascenso et al., 2020*) | Semantic segmentation | 89–95 | 87–93 | 86–94 | High (Atrous convolution) | Accurate segmentation with multi-scale feature extraction | Computationally expensive, memory-intensive |
| Autoencoder (*Nassif et al., 2021*) | Anomaly detection | 75–85 | 72–82 | 73–84 | Low (Feature extraction & reconstruction) | Unsupervised learning detects unknown defect types | Lower accuracy than supervised methods |

**Note:**
Accuracy: The proportion of correctly identified instances (both true positives and true negatives). Precision: The proportion of true positives among all instances identified as positive. Recall: The proportion of true positives among all actual positives. Computational Complexity: Refers to the amount of computational resources required, based on network depth, architecture, and processing requirements.

pre-trained networks on larger datasets like ImageNet (*Khanam et al., 2024*). Additionally, anomaly detection techniques, including autoencoders, identify defects by learning normal image features and highlighting deviations through reconstruction errors (*Hafiz & Bhat, 2020*). Advanced architectures like ResNet, DenseNet, and Inception handle variations in lighting, angles, and backgrounds, improving accuracy in diverse conditions (*Adhikari et al., 2024*). Furthermore, deep learning models such as recurrent neural networks (RNNs) and long short-term memories (LSTMs) analyze time-series sensor data to detect anomalies indicative of damage, while generative adversarial networks (GANs) augment training datasets with synthetic defect images, addressing data scarcity challenges (*Liu et al., 2021*). Despite requiring substantial computational resources and large labeled datasets, deep learning methods offer superior accuracy, robustness, and automation, making them highly effective for detecting complex defect patterns in wind turbines. Table 1 gives a comparison of the performance of different deep learning models used for detecting pollution and damage in wind turbines, based on key metrics such as accuracy, precision, recall, and computational complexity.

**Table 2 Object classes and numbers in the dataset.**

| Class | Number of objects |
| --- | --- |
| Paint damage | 851 |
| Erosion | 581 |
| Serration | 1,232 |
| Vortex | 221 |
| Vortex damage | 718 |

These models show varying strengths depending on the specific detection task, such as image classification, object detection, or segmentation. The choice of the model should be based on the accuracy and speed requirements, availability of labeled data, and computational resources.

# MATERIAL AND METHOD

Detection of damages in wind turbines from renewable energy sources is important for the life span of wind turbines. The detection of potential faults in turbines is essential for ensuring their optimal operation of turbines and predictive fault detection. In the study, a new method is presented for detection with deep learning models based on image processing. The results obtained with existing deep learning models are compared with the results obtained from the proposed deep learning model. The architectural structure of the proposed model, the dataset used, image collection processes, damage detection process, and object detection algorithm were clarified in this section.

## Dataset and pre-treatment

The Zeliha-04 dataset (*GTEK, 2024*) was utilized, containing a total of 1,883 images. These images include objects belonging to five classes: paint damage, erosion, serration, vortex, and vortex damage. In the dataset, the coordinates of each object are provided in a rectangular format. However, certain classes in the dataset included polygon-shaped coordinates, which were excluded from the dataset.

As a result, the number of images used in this study was reduced to 1,794. The classes and the number of objects in each class are presented in the accompanying table. The dataset consists of high-resolution images, each with dimensions of 1,080 × 1,920 pixels. Table 2 provides information on the types and numbers of damage in the images in the dataset.

Images belonging to the classes mentioned above were used within the study's scope. The number of objects in the classes provided sufficient diversity for the study and a suitable data set to test the models' success. The large number of classes in the data set was an important criterion for selection.

The images that make up the dataset were obtained with the DJI Matrice 300 RTK drone, an unmanned aerial vehicle. The specified unmanned aerial vehicle is an advanced drone model used especially in industrial and professional applications. It provides high-precision location determination. It is especially important in applications such as mapping and geographic information systems. Offering high-resolution camera

integration, this drone provides detailed images. It is widely used in industrial inspection, search and rescue operations, and research projects.

Data augmentation techniques were applied to improve model generalization and prevent overfitting due to the limited size of the dataset. Specifically, three augmentation strategies were used: (i) horizontal flip, (ii) vertical flip, and (iii) combined horizontal and vertical flip. Thus, the total number of images was increased by a factor of 4. These transformations support the model to learn damage features more robustly under various input perspectives by considering different camera angles and environmental conditions. The 1,794 images in the original dataset reached 7,176 images after data augmentation processes. During the data augmentation process, the following operations were performed on the images:

i. reflection on the horizontal axis,

ii. reflection on the vertical axis,

iii. reflection on both horizontal and vertical axes.

The dataset was augmented using horizontal reflection, vertical reflection, and combined reflections to enhance model robustness. Images were resized from 1,080 × 1,920 pixels to 840 × 840 pixels for consistency.

A specific transformation process was applied to make the coordinates of the objects in the images compatible. In this process, the coordinates were rearranged using the following equation: An equation that accounts for how a bounding box gave the coordinates of an object as x_min, y_min, x_max, y_max were arranged in a resized image when the image size changes are as follows. These transformations ensured that object bounding boxes remained proportional after resizing the images. The images describing this situation are given in Fig. 1. The original figure and the reproduced figures are given.

$$x' = x * \frac{w'}{w}. \tag{1}$$

$$y' = y * \frac{h'}{h}. \tag{2}$$

x,y: coordinates in the original image,
w,h: width and height of the original image,
x′,y′: new coordinates in the resized image,
w′,h′: width and height of the resized image.

Bounding box: it beconmes $x'_{min}, y'_{min}, x'_{max}, y'_{max}$.

To adjust the object coordinates for the resized image dimensions, we used scaling factors based on the ratio of the new size to the original size. For the x-coordinate, we multiplied the original value by (840/1,920), and for the y-coordinate, we used (840/1,080). For instance, an object at (960, 540) in the original image is mapped to (420, 420) in the resized image.

The categorical distribution and object numbers of the dataset after data augmentation are summarized in the table below. These pre-processing steps made the dataset more

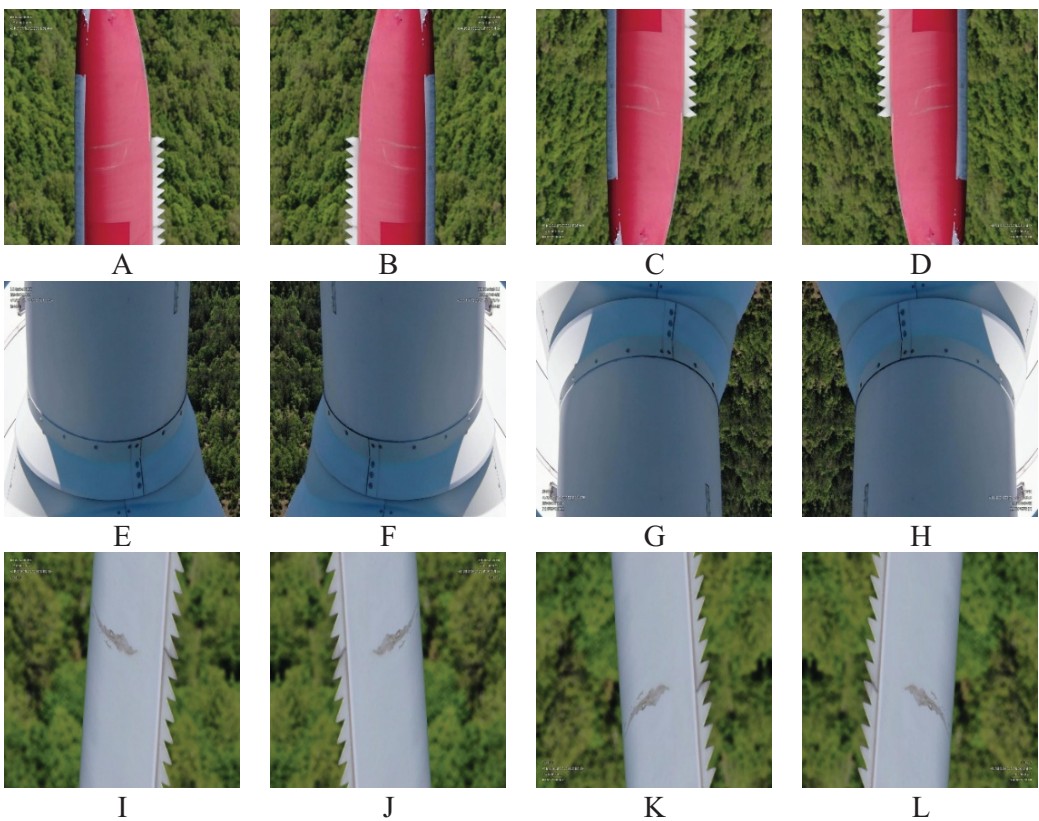

**Figure 1 Original images in the dataset and images obtained after data augmentation.** (A), (E), (I) the original images; (B), (F), (J) the reflections on the horizontal axis; (C), (G), (K) the reflections on the vertical axis; (D), (H), (L) the reflections on the horizontal and vertical axes.

suitable for deep learning models and increased the accuracy of the analysis process. Table 3 gives the number of objects belonging to each class after data augmentation.

Some images in the dataset and some images of the labeled coordinates in these images are shown in Fig. 2.

To ensure unbiased distribution of samples across training, validation, and test sets, a random sampling method was employed. This approach helps to prevent systematic selection bias by ensuring that images from different damage types and conditions are proportionally and randomly assigned to each subset. Additionally, to reduce the risk of overfitting and improve generalizability, a 5-fold cross-validation strategy was applied.

### Deep learning models

This article presents, 11 different deep learning models have been employed alongside the Faster R-CNN object detection algorithm to leverage their strengths and improve the overall detection process. Each of these deep learning models contributes in different ways, such as improving feature extraction, classification accuracy, or computational efficiency. For example, some models might be better suited for detecting specific types of objects, while others excel in handling large-scale datasets or real-time processing. By combining

**Table 3 Numbers of damage types in the dataset after data augmentation.**

| Class | Number of objects |
| --- | --- |
| Paint damage | 3,404 |
| Erosion | 2,324 |
| Serration | 4,928 |
| Vortex | 884 |
| Vortex damage | 2,872 |

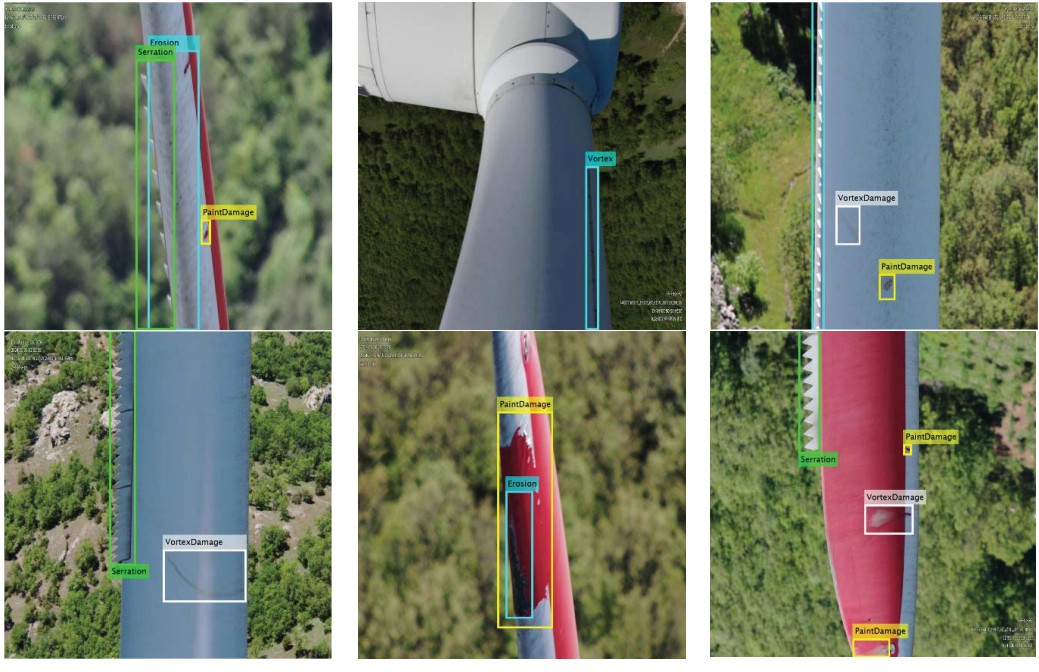

**Figure 2 Pattern examples in the dataset.**

Faster R-CNN with these models, the goal is to achieve better detection accuracy, faster processing times, and higher robustness in diverse environments.

### AlexNet

AlexNet (*Krizhevsky, Sutskever & Hinton, 2012*), which has five convolutional layers, three fully connected layers, and eight depths, consists of a total of 25 layers. This deep learning model is one of the pioneering deep learning models. It has become an important model with the different filters used and its single-branch structure. It consists of approximately 60 million parameters and 650 thousand neurons. Rectified Linear Unit (ReLU) activation function and Dropout layers were used for the first time in this model. It was selected due to its simple architecture and rapid training/testing capabilities.

### VGG16 ve VGG19

VGG16 (*Simonyan & Zisserman, 2014*) consists of 41 layers with 13 convolutional layers, three fully connected layers and a depth of 16. The VGG19 (*Simonyan & Zisserman, 2014*)

model has 19 depths, 16 convolutions, and three fully connected layers. It consists of 47 layers in total. There are 3 × 3 convolution filters in both models. It has an understandable model structure with its simple and deep structure. It is one of the most frequently used models in the literature. It is a model that is frequently preferred especially in image classification problems. It produced better results by extracting deeper features compared to AlexNet. It was selectede for this study due to its simplicity, fast training, efficient testing performance, strong feature extraction and being more successful in image classification processes.

### GoogleNet (InceptionV1)

This model has a depth of 22 and consists of a total of 144 layers. This model includes modules called inception. Convolution filters of different sizes are applied in parallel. The number of parameters is reduced with 1 × 1 filters. There are 170 connections to connect the layers. The inception module allows the model to contain fewer parameters. Thus, it takes up less space in memory. Thus, the computational cost is reduced. There are approximately 5 million parameters (*Szegedy et al., 2015*). The study was selected because it increases the performance of deep learning models and also has a low computational cost.

### ResNet (ResNet18, ResNet50, ResNet101)

Resnet18 (*He et al., 2016*) has a depth of 18 layers and consists of a total of 72 layers and 79 connections. This model has 11.7 million parameters. Resnet50 (*He et al., 2016*) has a depth of 50 layers and consists of a total of 177 layers and 192 connections. It has 25.6 million parameters. Resnet101 (*He et al., 2016*) has a depth of 101 layers and consists of 347 layers and 379 connections. It has 40 million parameters and the computational load is quite high compared to other models. It has been observed that the performance rates increase as the number of layers increases in Resnet models. Blocks called residual connections are used in these models. In such deep networks, it prevents gradient loss (vanishing gradient) and allows deeper models to be trained. In many studies, ResNet101 was chosen for its ability to address vanishing gradient issues, enabling deeper network training with superior accuracy in complex patterns.

### SqueezeNet

SqueezeNet (*Iandola, 2016*) is a model consisting of a total of 68 layers with a depth of 18 layers and 75 connections. This model includes the Fire module. 1 × 1 and 3 × 3 kernel filters are used in the Fire module. It contains 1.2 million parameters. This model comes to the fore in studies where a small number of parameters are needed. This model is used especially in environments with resource constraints. This model is more preferred in mobile and embedded systems.

### InceptionV3 ve InceptionResNetV2

InceptionResNetV2 (*Szegedy et al., 2017*) consists of a total of 825 layers and 922 connections with a depth of 164 layers. It is a model with inception and residual blocks. It has a deep structure with 55.9 million parameters. It is a very successful deep learning

model in obtaining features thanks to its depth. InceptionV3 (*Szegedy et al., 2016*) has a depth of 48 layers. It contains a total of 316 layers and 350 connections. It consists of 23.8 million parameters. This model includes the improved version of inception modules. Both models provide very successful performance in complex image processing problems. It was also selected to be presented in this article because it is better in large datasets.

### MobileNetV2

MobileNetV2 (*Sandler et al., 2018*) is a deep learning model developed specifically for mobile devices. It has a depth of 23 layers. It consists of a total of 153 layers and 162 connections. It contains 3.5 million parameters. This model includes inverted residual blocks. It is preferred in applications that require fast efficiency. Low hardware requirements can also be shown as one of the reasons for preference. It is also a preferable model for real-time applications.

Each deep learning model used in this study was chosen due to its advantages in addressing certain aspects of image classification or object detection. AlexNet was chosen due to its simplicity and fast training time, which makes it suitable as a baseline model. VGG16 and VGG19 with deeper architectures were included due to their strong feature extraction capabilities. ResNet models address the vanishing gradient problem. They were chosen to explore the effect of depth on accuracy. MobileNetV2 and SqueezeNet were chosen due to their lightweight architecture. Inception models and GoogleNet were chosen due to the effectiveness of the inception module in feature extraction and parameter reduction.

Each model was included in this study due to the most frequently encountered models in literature reviews and the reasons stated above. The combination of these models allows us to evaluate the trade-offs between accuracy, computational cost, and generalization. For instance, while ResNet101 provides high accuracy, its resource requirements are substantial. In contrast, SqueezeNet and MobileNetV2 offer faster inference and smaller memory footprints, which are critical for edge devices. This diversity enables a comparative analysis to identify the most appropriate model for real-world wind turbine applications.

### Recommended Model SatNET

SatNET (*Doğan, 2021*) is a deep learning model prepared for object detection from satellite images. It is a model developed for the detection of objects of different sizes on the ground surface from images obtained by satellite or remote sensing. This model consists of 115 layers and 131 connections, has a depth of 43 layers, and approximately 1.6 million parameters. The model is not pre-trained and is designed from scratch. Therefore, it does not have weights. A detailed image of the layered structure of the SatNET deep learning model is shown in Fig. 3.

The SatNET model has an input size of 248 × 248. The model consists of 12 blocks and each block has its own features. The first three blocks have convolution layers consisting of 1 × 1, 3 × 3 and 5 × 5 convolutions. In the first two blocks, the activation function is ReLU after the convolution layer, and in the 3rd block, the batch normalization layer is after the

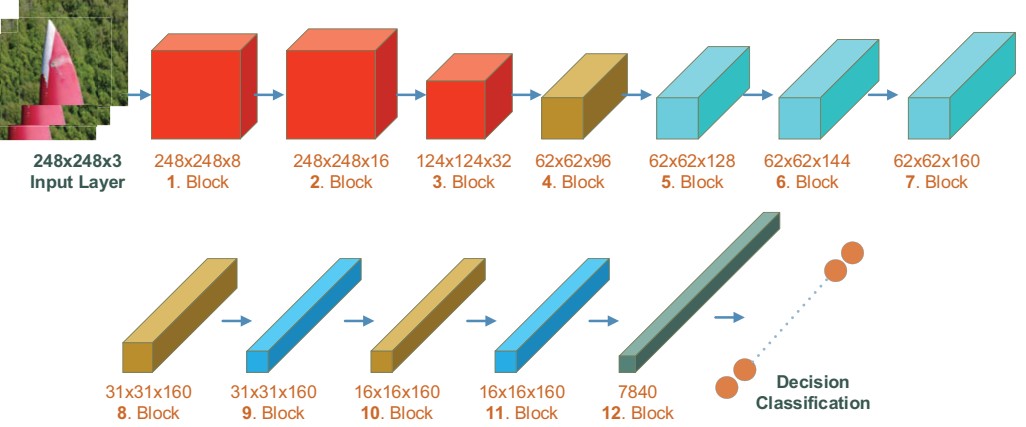

**Figure 3 Layered structure of SatNET deep learning model.**

convolution layer. In the 3rd block, the image size is reduced to $124 \times 124 \times 3$. In blocks 4, 8, and 10, the activation function is ReLU after the $3 \times 3$ convolution layer, and then the batch normalization layer. In the first layer of block 4, there is an additional activation function. In blocks 4, 8 and 10, the input size is reduced by half. In blocks 5, 6 and 7, there is a structure similar to the inception module. The blocks are similar. This block has a four branch internal structure and in the first branch, there is only a $1 \times 1$ convolution layer. In the second branch, there are three additional layers to the layer in the first branch. These are the activation function, $3 \times 3$ convolution layer, and Leaky ReLU activation layer respectively. In the third branch, instead of the Leaky ReLU activation function used in the second branch, there is ReLU, then the $3 \times 3$ convolution layer and then the Leaky ReLU activation layer. In the fourth branch, there is $1 \times 1$ convolution, batch normalization, activation, $3 \times 3$ convolution layer, batch normalization layer, and finally Leaky ReLU activation layer. In blocks numbered 9 and 11, there is a $3 \times 3$ convolution layer, cross-channel normalization layer, activation function, $3 \times 3$ convolution layer, cross-channel normalization layer, $3 \times 3$ convolution layer, and cross-channel normalization layer, respectively. In these blocks, the input size from the previous block remains constant. Block 12 is the block where the classification is made. This block includes first the activation function, dropout layer, average pooling layer, fully connected layer, softmax layer and finally the classification layer.

This model does not have weights because it is newly created. However, other models were taken as pre-trained. To evaluate the SatNET model under the same conditions as pre-trained, the model was trained twice. This approach shows that it would be more appropriate to compare it with other models.

Information about the features of the deep learning models used in this study and the reasons for their preference is given. The training of the models was taken as transfer learning, that is, pre-trained. Thus, it was subjected to fine-tuning for the damage detection task in wind turbines. The features of the deep learning models used are given in Table 4.

**Table 4 Features of deep learning models.**

| Architecture | Layers | Connections | Convolution Layers | Parameters | Top-1 error | Top-5 error |
|---|---|---|---|---|---|---|
| AlexNet (*Krizhevsky, Sutskever & Hinton, 2012*) | 25 | – | 8 | 62 m | 36.7 | 15.4 |
| VGG16 (*Simonyan & Zisserman, 2014*) | 41 | – | 16 | 138 m | 25.6 | 8.1 |
| VGG19 (*Simonyan & Zisserman, 2014*) | 47 | – | 19 | 144 m | 25.5 | 8 |
| GoogleNet (*Szegedy et al., 2015*) | 144 | 170 | 22 | 5 m | – | 6.67 |
| Resnet18 (*He et al., 2016*) | 72 | 79 | 18 | 11.7 m | 30.43 | 10.76 |
| ResNet50 (*He et al., 2016*) | 177 | 192 | 50 | 25.6 m | 22.8 | 6.71 |
| Resnet101 (*He et al., 2016*) | 347 | 379 | 101 | 40 m | 21.75 | 6.05 |
| SqueezeNet (*Iandola, 2016*) | 68 | 75 | 18 | 1.2 m | 41.90 | 19.58 |
| InceptionResnetv2 (*Szegedy et al., 2017*) | 825 | 922 | 164 | 55.9 m | 19.9 | 4.9 |
| Inceptionv3 (*Szegedy et al., 2016*) | 316 | 350 | 48 | 23.8 m | 21.2 | 5.6 |
| MobilNetV2 (*Sandler et al., 2018*) | 153 | 162 | 23 | 3.5 m | 28.0 | 8.6 |
| SatNET | 115 | 131 | 43 | 1.6 m | – | – |

**Table 5 Hyperparameters used for training deep learning models.**

| Hyperparameter | Value | Explanation |
|---|---|---|
| Optimizer | SGDM (Stochastic Gradient Descent with Momentum) | Optimization algorithm used. |
| LearnRateSchedule | Piecewise | The learning rate is reduced in parts over time. |
| LearnRateDropFactor | 0.1 | The decay factor of the learning rate (multiplied by 0.1 at each step). |
| LearnRateDropPeriod | 1 | Specifies every number of epochs that the learning rate was decreased. |
| InitialLearnRate | 1e−4 (0.0001) | Initial learning rate at the beginning of training. |
| MaxEpochs | 5 | Maximum number of epochs (how many times to go through the dataset). |
| MiniBatchSize | 2–16 | The number of data samples to use in an iteration. |
| Verbose | True | Whether to show detailed output during training. |
| ValidationFrequency | 20 | How many iterations were required to check a validation dataset? |

All models were trained using the same hyperparameters on the dataset containing wind turbine damage. Table 5 shows the hyperparameter values used for training the models. Different values were used due to the change in the number of parameters of the minibatch size models. The values vary between 2–16.

## Faster R-CNN for object detection

Object detection algorithms are not typically deployed alone in real-world applications. They are often integrated with various deep learning models to enhance their capabilities and improve performance. Object detection is a complex task that involves not only identifying objects within images but also accurately classifying and localizing them. Deep learning models, particularly CNNs, are widely used in this context to process and analyze image data more effectively.

There are different object detection algorithms used in literature to detect the object in the image. YOLO (*Redmon, 2016*) algorithm is released with different versions almost

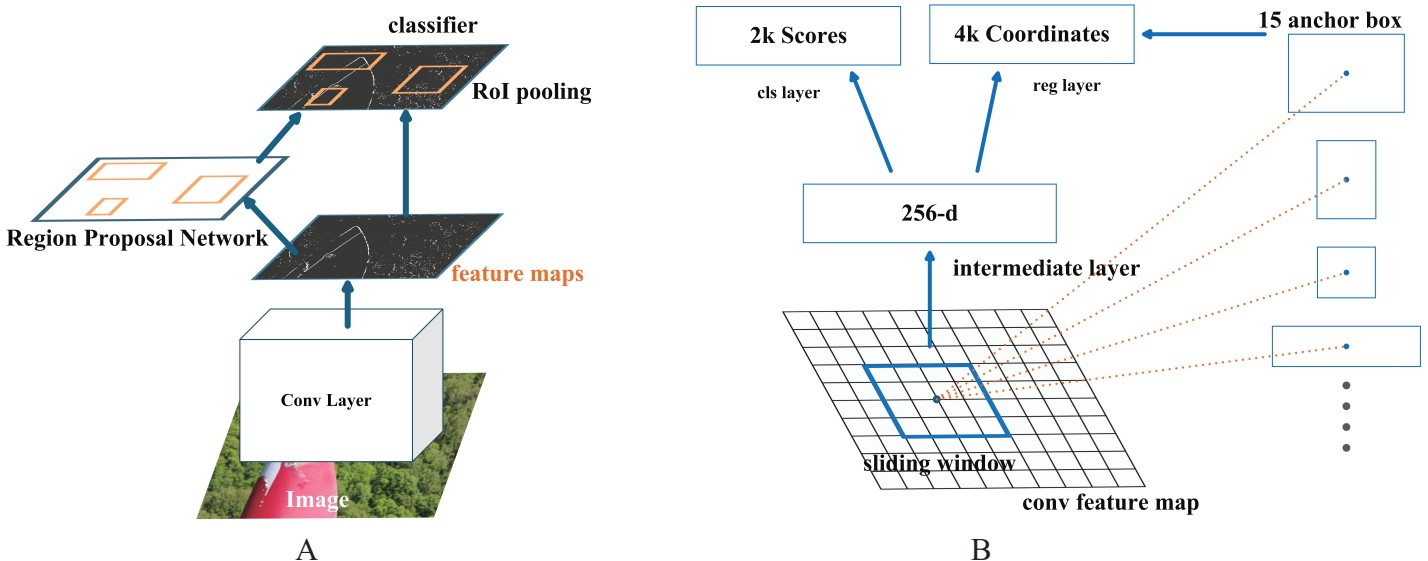

**Figure 4** (A) Region Proposal Network (RPN) used in the study (B) Working logic of R-CNN object detection algorithm.

every year. YOLO is one of the object detection algorithms used in literature. Single Shot MultiBox Detector (SSD) (*Liu et al., 2016*) tries to detect the object in a single step. It is quite fast but it is behind YOLO and Faster R-CNN (*Ren et al., 2016*) in terms of performance rate. RetinaNet (*Lin, 2017*) is an algorithm used for detecting small objects. EfficientDet (*Tan, Pang & Le, 2020*) is an efficiency-oriented object detection algorithm offered by Google. In this study, the Faster R-CNN (*Ren et al., 2016*) object detection algorithm was used to detect the damage and damage types on the wind turbine blades. This algorithm provides high accuracy and exhibits a 2-stage approach. The first of these stages is region proposal, and the second is object classification and bounding box regression.

Region Proposal Network (RPN) is a convolutional network. It uses anchor boxes to determine possible object regions in the image. The anchor box used to determine the region has different sizes and ratios. In this study, the number of anchor boxes of different sizes used to determine the region is determined as 15. This algorithm, which makes region suggestions for object detection in the image, is converted into fixed-size feature maps. This process is carried out by the region of interest (ROI) pooling layer. Feature maps are used for object classification and bounding box regression. This situation is shown in Fig. 4.

Faster R-CNN is used with deep learning models to detect damage and types in wind turbines. Each model is trained with the Faster R-CNN structure and the object is detected by classifying it with the suggested target boxes. The algorithm designed to detect the damage and its classes in the study is shown in Fig. 5. All models are run according to this flow.

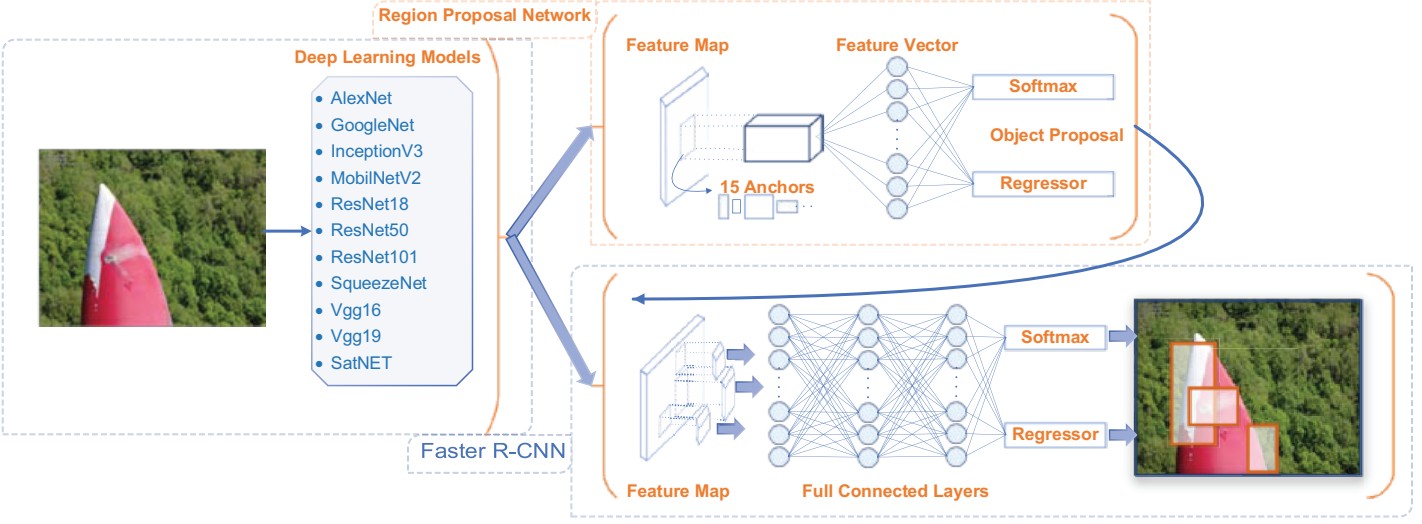

**Figure 5 Application algorithm for determining the type and type of damage used in the study.**

Faster R-CNN object detection algorithm was selected due to its high accuracy and robustness in detecting multiple object classes in complex scenes. Instead of single-stage detectors (*e.g.*, YOLO, SSD), Faster R-CNN adopts a two-stage approach that first generates region proposals and then classifies the bounding boxes. It provides more precise detection, which is especially important in wind turbine damage analysis, where defects (*e.g.*, erosion, paint damage, vortex) may be small, irregular, and visually similar. Faster R-CNN is well suited for the high-resolution image data used in this study and can effectively detect fine-grained damage types under varying light and weather conditions. Considering these advantages, Faster R-CNN is a strong candidate.

## Evaluation metrics

In object detection tasks, the evaluation of model performance is crucial to understanding how well the algorithm is identifying and localizing objects within an image. The most commonly used evaluation metrics are precision and recall, which measure the accuracy and completeness of the model's detections. Precision refers to the proportion of true positive detections (correctly identified objects) out of all the detections made by the model, including false positives (incorrect detections). Recall, on the other hand, measures the proportion of true positives out of all actual objects in the image, including false negatives (missed objects). To provide a comprehensive evaluation, the average precision (AP) is calculated, which is the area under the precision-recall (PR) curve. The PR curve is created by plotting precision against recall at different thresholds of detection confidence. The AP value summarizes the model's precision and recall performance across all possible detection thresholds, with a higher AP indicating better overall performance. The diagram below illustrates how precision and recall are calculated and how they are used to generate the AP score, offering a clearer understanding of the model's effectiveness in detecting objects.

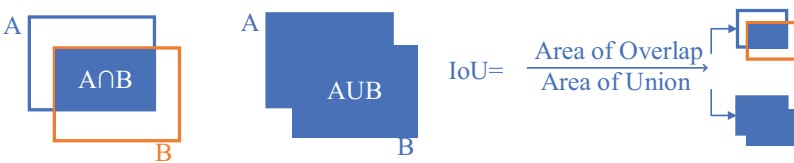

**Figure 6  IuO metric calculation.**               

**Non-Max Suppression (NMS)**: Non-Max Suppression is used to identify the correct bounding boxes in object detection. It retains bounding boxes with an Intersection Over Union (IoU) score greater than 0.5 and suppresses the others. If multiple bounding boxes with an IoU greater than 0.5 are detected for the same object, the one with the highest confidence score is selected and used.

**Intersection Over Union (IoU)**: Intersection Over Union (IoU) is a measure of accuracy for bounding boxes. It compares a predicted bounding box with the actual ground truth bounding box. The IoU value is determined by calculating the overlap between the predicted and actual boxes, divided by the area of their union. Figure 6 shows the IoU calculation process.

Originally, the R-CNN architecture was used for object detection, but it required classifying 2,000 region proposals per image, which made training the network very time-consuming. As a result, a faster architecture was developed, known as the Faster R-CNN algorithm, which improved the speed and efficiency of the detection process.

### Computing infrastructure

The experiments and deep learning model training were conducted on a high-performance computer running Windows 11 Pro as the operating system. The system was equipped with an Intel core i9-14900 processor, 48 GB of ram, a 3 TB ssd for storage, and an Nvidia Rtx 4080 graphics card with 16 GB of vram, which provided the computational power required for deep learning tasks. The implementation of the models was carried out using Matlab, specifically leveraging its built-in Deep Learning Toolbox for model design, training, and evaluation. All scripts and functions were developed and executed within the matlab environment, ensuring a streamlined workflow for the experiments.

## EXPERIMENTAL RESULTS

Wind turbines rely heavily on the performance of their generators and back-to-back (B2B) converters to ensure efficient energy conversion and grid compatibility (*Kasasbeh et al., 2020*; *Saygin & Aksöz, 2017*; *Yilmaz et al., 2018*). The generator transforms mechanical energy into electrical energy, while the B2B converter regulates variable frequency power into grid-compatible output. Faults in these components, such as insulation degradation in generators or insulated-gate bipolar transistor (IGBT) failures in converters, can lead to inefficiencies, energy losses, and grid disturbances. Deep learning models, trained on operational data such as voltage, current, rotor speed, and temperature, have proven effective in detecting these faults (*Wang et al., 2021*, *2020*). For instance, issues like

unbalanced magnetic pull or winding short circuits in generators, and thermal stress or harmonic distortions in converters, can be identified through pattern analysis in real-time data. Evaluating these models involves metrics like precision and recall, which measure the accuracy and completeness of fault detections (*Lan et al., 2020*; *Wang et al., 2021*). AP, calculated from the precision-recall (PR) curve, offers a comprehensive measure of performance, while techniques like Non-Max Suppression (NMS) and Intersection Over Union (IoU) enhance fault localization. Faults in generators or B2B converters can have significant grid impacts, including harmonics, voltage sags, and reactive power imbalances, potentially compromising grid stability (*Lan et al., 2020*). By integrating deep learning-based fault detection with grid simulation tools, operators can predict and mitigate these effects proactively. Advanced models like Faster R-CNN enable real-time fault diagnosis, ensuring reliable turbine operation and minimizing disruptions to the power system (*Saygin, Aksoz & Yilmaz, 2016*).

In this study, an analysis was performed for damage detection using 11 different deep learning models and the Faster R-CNN object detection algorithm. 10 deep learning models frequently used in the literature were evaluated as pre-trained, and a new model, the SatNET, was presented and its performance was compared with other models. The evaluation focused on damage types and memory space. The advantages and performance of the new model, the SatNET, were evaluated. The performance of the models according to damage types is given in Table 6. The AP value for each damage type is given in the table.

Model Performances by Damage Types:

In paint damage detection, ResNet50 (58.4%) and SatNET achieved high accuracy for this damage category with a success rate of 55.7%. AlexNet (9.8%) and ResNet18 (3.6%) provided accuracy and exhibited the lowest performance among deep learning models. The performance of ResNet50 and SatNET is quite close and considering the space it occupies in memory, it is clear that the SatNET has the advantage.

In erosion damage detection, ResNet101 (80.3%), SatNET, 76.7%, and VGG19 (75.8%) showed good performance in this damage with accuracy rates. When we look at ResNet101, SatNET, and VGG19 models, SatNET's performance with its lightweight structure is quite remarkable.

In serration detection, VGG19 (95.4%) and SatNET, with 95.2% accuracy rates, were the best-performing models in this damage type. These two models showed almost the same performance. The SatNET is notable for its lower memory requirements.

In vortex detection, ResNet50 (93.1%) and ResNet101 (87.5%) provided the highest performance in this type with their performances. The SatNET, on the other hand, showed a moderate performance with a 66.1% accuracy rate. Although it lags other models, this performance is considered to be at an acceptable level when the SatNET's lightweight structure and low resource consumption are considered.

In vortex damage detection, ResNet101 (51.5%), ResNet50 (46.4%), and SatNET show the 3 deep learning models that perform best with 27.3% accuracy. It is seen that the SatNET provides above-average performance in this damage type, whereas other models

**Table 6 AP values obtained from deep learning models according to damage types.**

| | Paint damage | Erosion | Serration | Vortex | Vortex damage |
|---|---|---|---|---|---|
| AlexNet | 0.098 | 0.572 | 0.899 | 0.166 | 0.044 |
| GoogleNet | 0.280 | 0.713 | 0.930 | 0.186 | 0.076 |
| InceptionV3 | 0.321 | 0.740 | 0.934 | 0.694 | 0.155 |
| MobilNetV2 | 0.295 | 0.696 | 0.929 | 0.412 | 0.056 |
| ResNet18 | 0.036 | 0.664 | 0.893 | 0.254 | 0.011 |
| ResNet50 | 0.584 | 0.787 | 0.944 | 0.931 | 0.464 |
| ResNet101 | 0.541 | 0.803 | 0.936 | 0.875 | 0.515 |
| SequeezeNet | 0.327 | 0.706 | 0.921 | 0.749 | 0.120 |
| VGG16 | 0.425 | 0.753 | 0.939 | 0.722 | 0.231 |
| VGG19 | 0.439 | 0.758 | 0.954 | 0.481 | 0.240 |
| SatNET | 0.557 | 0.767 | 0.952 | 0.661 | 0.273 |
| Average | 0.355 | 0.724 | 0.930 | 0.557 | 0.199 |

**Table 7 Memory usage, total test time and fps values of the models.**

| | Space occupied in memory | Total testing time (s) | Frames per second |
|---|---|---|---|
| AlexNet | 824 MB | 48.4 | 14.83 |
| GoogleNet | 266 MB | 163.3 | 4.40 |
| InceptionV3 | 461 MB | 324.2 | 2.21 |
| MobilNetV2 | 230 MB | 150.6 | 4.77 |
| ResNet18 | 311 MB | 51.3 | 14.00 |
| ResNet50 | 474 MB | 179.8 | 3.99 |
| ResNet101 | 705 MB | 191.9 | 3.74 |
| SequeezeNet | 201 MB | 97.3 | 7.38 |
| VGG16 | 1,674 MB | 72.39 | 9.92 |
| VGG19 | 1,733 MB | 80.2 | 8.95 |
| SatNET | 192 MB | 46.5 | 15.44 |

also show low performance. When all models and damage types are considered, the SatNET model has performed above average and shows a successful performance.

Considering the space it takes up in memory, the SatNET has become the model that requires the least memory with only 192 MB. This provides a significant advantage in situations where resource needs are limited. When compared to other models, it has been observed that the models that require the most resources are the VGG16 and VGG19 models. The SatNET stands out in environments where resource constraints and high performance is desired and shows that it is an important option. In addition, SatNET completes the test process in the dataset of the models in a short time. Information on how long (in seconds) the 10% part allocated for the test in the dataset is completed and the memory usage areas of the models are shown in Table 7. The number of images processed per second is also given in the table.

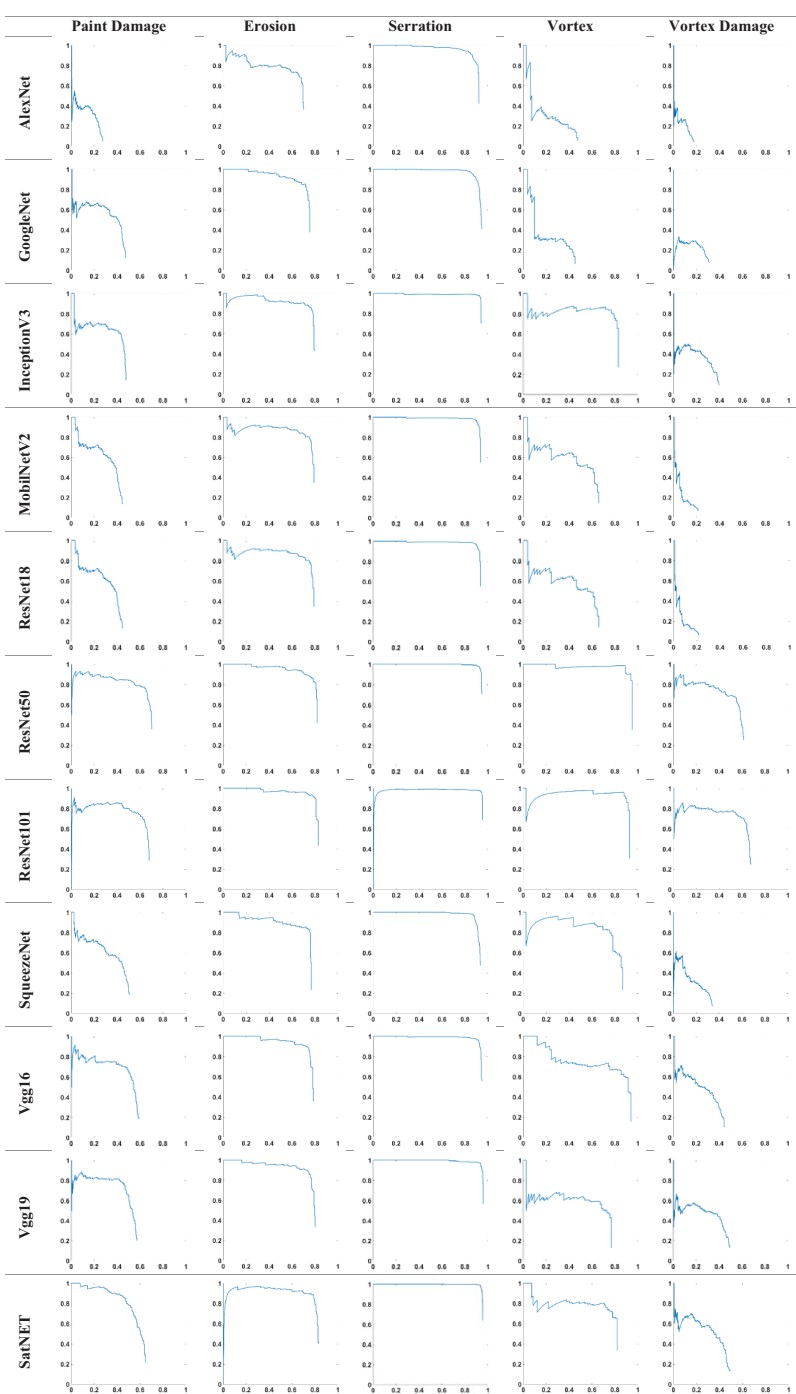

**Figure 7 Comparison of average precision values obtained from all deep learning models in five different wind turbine damage types (paint damage, erosion, serration, vortex and vortex damage).**

AP graphs of the results obtained from deep learning models are given in Fig. 7. A comparative graph of the turbine damage and classes of each model is presented. Figure 7 compares the average precision (AP) values across different models and damage types, highlighting SatNET's efficiency in memory-limited environments. Each graph represents the AP value achieved by a model for its corresponding damage type. It highlights the performance variability between models and helps determine which architectures are better suited to detect certain damage types.

The proposed model demonstrates high efficiency in damage detection, owing to its superior performance and low resource consumption. This provides a balanced solution. In addition, low memory usage provides a great advantage in terms of real-time applications. The speed of the model and low requirements makes the model stand out. The generalization ability of the SatNET model has shown that it can be used in different applications.

## CONCLUSION AND FUTURE WORKS

This study has demonstrated the significant potential of deep learning models, particularly the proposed SatNET model, in detecting and classifying damage types in wind turbines with high accuracy and low resource consumption. The SatNET's lightweight design and minimal memory requirements position it as an optimal solution for real-time and resource-constrained applications. The results of the study highlight the model's competitive performance across various damage categories, validating its suitability for practical deployment in wind turbine maintenance and monitoring. Furthermore, the integration of advanced data augmentation techniques and a carefully constructed datasethas ensured the robustness and generalizability of the proposed framework. Thus, it is suitable for deployment in edge devices such as drones or embedded systems used in wind turbine inspections, making it applicable for real-time damage detection during aerial surveillance without requiring high computational cost and high GPUs.

Despite the favorable results, several areas for improvement and future research remain. Enhancing the environmental adaptability of SatNET is a critical next step. Training the model under a broader range of conditions, such as varying lighting, weather, and operational states, will ensure more robust and reliable performance in real-world scenarios. Additionally, integrating SatNET model with advanced sensing systems, such as real-time data collection *via* drone inspections or sensor networks, can significantly enhance its detection capabilities. Such integration would not only improve the precision of damage identification but also enable scalable applications for monitoring large wind farms. However, there are limitations of the study. One of these limitations is that the dataset is limited to five damage classes. Another is that while SatNET is efficient, its accuracy for some damage types (*e.g.*, vortex damage) is lower compared to heavier models such as ResNet101. The low performance in detecting vortex damage (27.3% AP) should be analyzed. One of the possible reasons for this damage type is data imbalance. The number of annotated vortex damage images in the dataset is low compared to other classes. Also, vortex damage has similar visual features to other damage types such as erosion or vortex. This makes class separation difficult. The lightweight architecture of SatNET may

limit its capacity to learn the fine distinctions required to reliably detect this damage type. Among the future improvement works, it would be more appropriate to collect more labeled vortex damage data. In addition, techniques such as focus loss or attention mechanisms can be considered to improve class separation.

Expanding the scope of the SatNET model to other renewable energy systems, such as solar panel or hydroelectric infrastructure inspections, represents another exciting direction. The model's architecture can be adapted and fine-tuned to address diverse challenges in renewable energy maintenance, further broadening its impact. Moreover, future work will focus on optimizing the model through techniques such as pruning and quantization to reduce computational overhead while maintaining or even enhancing its accuracy.

Another potential research direction involves the integration of predictive analytics into SatNET's framework. By incorporating anomaly detection and prognostic capabilities, the system could provide valuable insights into the progression of damage over time. This would facilitate predictive maintenance, minimize operational downtime, and ultimately reduce costs. Finally, extensive field testing and real-world deployment are essential to validate the model's performance under diverse operational conditions and to gather feedback for further refinement. Future work will focus on addressing these limitations by including a wider set of real-world damage types and environmental conditions in the training data. In addition, integrating temporal data or video sequences can increase model robustness. We also plan to explore model compression techniques such as pruning and quantization to further reduce the computational burden for embedded deployment. Field testing of the model on active turbines will be important to verify its real-world performance and reliability. In the study, only Faster R-CNN was used among object detection methods. In subsequent studies, it is considered to use YOLO versions of single-stage detection methods.

In conclusion, this study provides a foundational framework for automated wind turbine monitoring using deep learning models. By addressing the outlined future objectives, SatNET can evolve into a comprehensive, scalable, and efficient solution, contributing to the optimization of renewable energy systems and supporting global sustainability goals. In addition, its low memory requirement and lightweight structure make SatNET suitable for platforms such as NVIDIA Jetson Nano, Jetson Xavier or Google Coral used in unmanned aerial vehicle (UAV) systems. It can run efficiently on embedded processors with a speed of 15.44 fps. In order to provide faster performance in SatNET's UAV integration, it can be exported to edge compatible formats such as Open Neural Network Exchange (ONNX) or TensorRT. Thus, real-time damage detection is possible. It reduces the need for manual processing and increases operational efficiency.

### Funding

This article is supported by the European Union's Horizon Europe research and innovation program under grant agreement No. 101084323, project BLOW (Black Sea

fLoating Offshore Wind). The funders had no role in study design, data collection and analysis, decision to publish, or preparation of the manuscript.

## Grant Disclosures
The following grant information was disclosed by the authors:
European Union's Horizon Europe Research and Innovation Program: 101084323.

## Competing Interests
The authors declare that they have no competing interests.

## Author Contributions
- Ferdi Doğan conceived and designed the experiments, performed the experiments, analyzed the data, performed the computation work, prepared figures and/or tables, authored or reviewed drafts of the article, and approved the final draft.
- Saadin Oyucu conceived and designed the experiments, analyzed the data, prepared figures and/or tables, authored or reviewed drafts of the article, and approved the final draft.
- Emre Bicer analyzed the data, prepared figures and/or tables, authored or reviewed drafts of the article, and approved the final draft.
- Ahmet Aksoz analyzed the data, prepared figures and/or tables, authored or reviewed drafts of the article, and approved the final draft.

## Data Availability
The Zeliha-T04 Computer Vision Dataset is available at https://universe.roboflow.com/gtek/zeliha-t04.

The data is available at Zenodo: doganferdi. (2025). doganferdi/windturbinedefectdetection: windturbine damage detection (windturbine). Zenodo. https://doi.org/10.5281/zenodo.15609706.

## Supplemental Information
Supplemental information for this article can be found online at http://dx.doi.org/10.7717/peerj-cs.3163#supplemental-information.

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
