# Peer review of "Deep learning models for damage type detection in wind turbines"

_PeerJ Computer Science, doi:10.7717/peerj-cs.3163_

## Round 0.1 · original submission · Major Revisions

· Academic Editor

Major Revisions

**Language Note:** PeerJ staff have identified that the English language needs to be improved. When you prepare your next revision, please either (i) have a colleague who is proficient in English and familiar with the subject matter review your manuscript, or (ii) contact a professional editing service to review your manuscript. PeerJ can provide language editing services - you can contact us at [email protected] for pricing (be sure to provide your manuscript number and title). – PeerJ Staff

Reviewer 1 ·

Basic reporting

well structured in general

Experimental design

Method presents a description, but the novelty of the paper can be better highlighted and valorized

Validity of the findings

-

Additional comments

1. The abstract currently uses somewhat promotional language ("cutting-edge deep learning techniques") and lacks explicit quantitative results or precise findings. The authors should rewrite the abstract to include clear, specific outcomes, highlighting numerical values or key metrics (e.g., accuracy improvements or computational performance).

2. The manuscript frequently employs informal and subjective language (e.g., "leveraged," "meticulously curated," "promising avenue," "strong foundation"). Such expressions detract from the scientific tone required by an academic journal. Please revise these sections using neutral, precise scientific language.

3. The use of technical terminology, especially related to deep learning methods, is inconsistent. For example, activation functions are labeled inconsistently ("relu," "leakage activation"). Standardize these to common terms widely recognized in machine learning literature (e.g., "ReLU," "Leaky ReLU") throughout the manuscript.

4. The manuscript contains significant formatting errors, particularly incorrect section numbering such as "371. Introduction" and "1102. Literature Review." This inconsistency must be corrected to sequential numbering (1, 2, 3, etc.) to conform with academic standards.

5. Statements like "minimize selection bias" when describing random allocation of images are unclear without proper explanation or justification. Clarify explicitly how random assignment helps ensure unbiased results or adjust wording to reflect accurately the method's intent and scientific rationale.

6. The manuscript suffers from repetitive wording, including the title itself ("Damage and Damage Types"). This redundancy reduces readability and clarity. A suggested concise revision for the title is "Deep Learning Models for Damage Type Detection in Wind Turbines." Also, phrases such as "damage and contamination" vs "damage and contaminations" should be standardized and clarified.

7. Certain technical descriptions lack necessary clarity and detail. For instance, the description of data augmentation and bounding-box coordinate adjustments (equations provided without sufficient context) needs to be expanded with clear explanations to ensure reproducibility by readers unfamiliar with these specific transformations.

8. Throughout the manuscript, there are grammatical and punctuation errors, notably missing spaces after periods and occasional double periods (e.g., lines 18-21 in the Introduction). Thorough proofreading and editing are necessary to rectify these basic linguistic issues to enhance readability.

9. The manuscript introduces multiple deep learning architectures (Sections 3.2 and 3.3) but lacks a clear explanation of why specific models were chosen and how each contributes uniquely to addressing the stated problem. The authors should explicitly discuss interconnections among selected methods, clarify their respective roles, and justify their relevance to wind turbine damage detection.

10. Figures (e.g., Figure 7 AP graphs) require clearer, more descriptive captions and legends to help readers fully understand what is depicted without referring excessively back to the text. A brief explanation within the caption should clearly state what readers should interpret from each figure.

11. The discussion and conclusion sections should more explicitly highlight practical implications, limitations, and realistic deployment scenarios of the proposed deep learning models. Furthermore, proposed future research directions should be clearly linked to specific limitations or gaps identified within the current study.

Cite this review as

·

Basic reporting

The manuscript is well written in professional, academic English. The background provides a strong motivation for the study, emphasizing the importance of damage detection in wind turbines using deep learning models. The figures and tables are generally clear, informative, and support the manuscript's flow. However, I suggest the following improvements:
1) Improve clarity in Figure 7 by standardizing color codes and adding a legend that clearly maps models to colors for easier interpretation.
2) Consider reorganizing Table 6 to make the best-performing models for each damage type more visually apparent (e.g., bolding or highlighting top AP values).
3) Include a brief paragraph in the introduction or literature review section summarizing the specific gaps in existing CNN-based methods that SatNET aims to fill.

Experimental design

The authors present a solid experimental design using a well-curated, annotated dataset of 1,794 images, expanded through augmentation to 7,176 samples. The use of 5-fold cross-validation is commendable and strengthens the generalizability of the findings.
However, the following points need attention:
1) The manuscript exclusively uses Faster R-CNN for object detection. Incorporating or comparing with single-stage detectors (e.g., YOLOv5/YOLOv8) would provide a more complete benchmark.
2) The authors mention memory efficiency of SatNET but do not provide inference time or FPS (frames per second). These are critical for real-time deployment claims.
3) Data augmentation strategies are outlined, but the specific augmentation combinations and randomization process should be explained more precisely.

Validity of the findings

The SatNET model demonstrates promising results with competitive AP scores and notably low memory usage (192MB), validating its efficiency in resource-constrained scenarios. The comparison with established pre-trained CNNs is thorough. The low performance in detecting Vortex Damage (27.3% AP) should be more thoroughly discussed. The authors should explore whether this is due to data imbalance, visual similarity to other defects, or architecture limitations.
The manuscript does not report mAP@50, mAP@95, or IoU per damage class, which are standard metrics in object detection. Including these would enhance the comparability of results with other literature.
The proposed model was tested under ideal imaging conditions. Generalization to diverse environmental conditions (e.g., lighting, rain, motion blur) should be explored or mentioned as a limitation.

Additional comments

The proposed SatNET architecture is a contribution to lightweight damage detection frameworks. It provides a practical balance between accuracy and memory consumption, making it suitable for real-time applications on drones or embedded systems.
Additional suggestions for improving the manuscript:
Include a mini ablation study or justification for design choices in SatNET (e.g., use of inception-like modules, activation functions, normalization layers).
Discuss how SatNET could be integrated with UAV systems or edge AI platforms for field deployment.
Future work could explore model compression techniques (quantization, pruning) and cross-domain adaptability (e.g., to solar panel or pipeline inspections).
Proofread the manuscript to fix grammar and punctuation errors throughout

Cite this review as

---

## Round 0.2 · accepted · Accept

· Academic Editor

Accept

Dear Author,
Your paper has been accepted for publication in PEERJ Computer Science. Thank you for you fine contribution.